# On the Ambiguity in Classification

**Arif Dönmez**\*                    ARIF.DOENMEZ@IUF-DUESSELDORF.DE

*IUF - Leibniz Research Institute for Environmental Medicine*

**Editors:** Sophia Sanborn, Christian Shewmake, Simone Azeglio, Arianna Di Bernardo, Nina Miolane

## Abstract

We develop a theoretical framework for geometric deep learning that incorporates ambiguous data in learning tasks. This framework uncovers deep connections between noncommutative geometry and learning tasks. Namely, it turns out that learning tasks naturally arise from groupoids, and vice versa. We also find that learning tasks are closely linked to the geometry of its groupoid $*$-algebras. This point of view allows us to answer the question of what actually constitutes a classification problem and link unsupervised learning tasks to random walks on the second groupoid cohomology of its groupoid.

**Keywords:** Geometric Deep Learning, Noncommutative Geometry, Unsupervised Learning, Learning Tasks, Classification Problem, Ambiguous Data

## 1. Introduction

In this paper, we develop a theoretical framework for geometric deep learning that incorporates ambiguous data in learning tasks. Thinking of smoothly morphing an image into another image or speech-to-text problems lets us conclude that real-world applications are full of ambiguous data. As far as we could survey an approach that incorporates ambiguous data does not exist yet. For this, we start studying the approaches from different areas in mathematics that tackle classification problems. Naturally in these contexts, the study of group actions arises as a model for classification problems. We start with studying approaches in the algebraic setting (Section 3). As an ex-curs into the application-driven deep learning approach (Section 4) suggests to us, we continue further to the harmonic analysis imprinted approaches (Section 5). We find out that in both the algebraic and the harmonic analysis setting, the representation theory of the regular representation of the group algebra plays a central role. By these investigations we discover a new class of functions, so-called *expected-to-be-invariant* functions (Definition 8) which generalizes the notion of *invariant* functions and seems better suited for applications. So far all classification problems were induced by group action. In particular, we did not deal with ambiguous data yet. In Section 6 we find out that we can capture the distribution of the data from the sample space with respect to the group action via so-called Radon-Nikodym cocycles. After that, we enter into the groupoid setting. Groupoids allow us to handle ambiguous data. According to our previous observations in the algebraic and harmonic analysis setting, we focus on groupoid algebra and its regular representation. For locally compact Hausdorff groupoid with left Haar system, we introduce its twisted groupoid $*$-algebra following Renault (2006) in Section 7. After these steps we are finally able to answer the question of what actually constitutes a classification problem/learning task:

---

\* www.arifdoenmez.github.io

**Definition 13** A classification problem consists of a locally compact, Hausdorff groupoid $\mathcal{G}$ with a left Haar system $\{\lambda^u\}_{u \in \mathcal{G}_0}$, and a normalized continuous $\mathbb{T}$-valued 2-cocycle $\sigma$ on $\mathcal{G}$. The space $\mathcal{G}_0$ is called *sample space*, and $\sigma$ is called *labeling*.

This new viewpoint allows us to link unsupervised learning to random walks on the second groupoid cohomology with values in $\mathbb{T}$, the multiplicative group of complex numbers of modulus 1.

## 2. Definitions and Notation

At first, we need to introduce a generic definition of a classification problem:

**Definition 1** *A classification problem* $\mathsf{C}$ *is a tuple* $(\Omega, (\Omega_l)_{l \in L})$ *consisting of a set* $\Omega$*, so-called sample space, and family* $(\Omega_l)_{l \in L}$ *of subsets of* $\Omega$ *such that*

*a)* $\Omega = \bigcup_{l \in L} \Omega_l$,

*b)* $\bigcup_{l \in L'} \Omega_l \subsetneq \Omega$ *for all* $L' \subsetneq L$.

*Thereby, a subset* $\Omega_l$ *($l \in L$) is called cluster and the index set* $L$ *is called the set of classes of* $\mathsf{C}$.

In this note, we will distinguish between two types of classification problems:

**Definition 2** *We call a classification problem* $\mathsf{C}$ *unambiguous if its clusters are pairwise disjoint. Otherwise, we call it ambiguous.*

**Definition 3** *A classification problem* $\mathsf{C}$ *is induced by a group action if there is a group* $G$ *acting on its sample-space* $\Omega$ *such that the clusters are given by the orbits of the* $G$ *action.*

**Remark 4** *For a classification problem* $\mathsf{C} = (\Omega, (\Omega_l)_{l \in L})$ *induced by a group action* $G : \Omega$, *there is an one-to-one relation between the index set* $L$ *and* $\Omega/G$ *(in the set-theoretical sense).*

**Lemma 5** *A classification problem induced by a group action is an unambiguous classification problem, and vice versa.*

## 3. Algebraic classifications

For all notions concering basic algebraic geometry and invariant theory we refer to Shafarevich and Reid (1994) and Kraft and Wiedemann (1985). For this section, we fix an algebraically closed field $\mathbb{K}$, and for $n \in \mathbb{N}$ let $\mathbb{A}^n$ denote the affine variety $\mathbb{K}^n$. All algebraic varieties in this section will be over $\mathbb{K}$.

Let $\mathsf{C} = (\Omega, (\Omega_l)_{l \in L})$ be a classification problem with an affine variety $\Omega$ as sample space. Further assume that $\mathsf{C}$ is induced by an algebraic group $G$ acting regular on $\Omega$. Then, the clusters are locally closed subsets of the sample space and we have an associated unambiguous classification problem $\bar{\mathsf{C}} = (\Omega, (\bar{\Omega}_l)_{l \in \bar{L}})$ where the clusters are induced by the closure of the orbits. Regarding the classification problem in this setting, it is natural to consider first the *associated regular $G$-representation* $\mathbb{K}[\Omega] = \{f : \Omega \to \mathbb{K} \mid f \text{ is regular}\}$

given by $(g.f)(\omega) := f(g^{-1}\omega)$, $g \in G$, $f \in \mathbb{K}[\Omega]$, $\omega \in \Omega$ (Kraft and Wiedemann (1985)), then the *invariant ring* $\mathbb{K}[\Omega]^G := \{f \in \mathbb{K}[\Omega] \ : \ g.f = f \text{ for all } g \in G\} \subseteq \mathbb{K}[\Omega]$ and focusing on the challenge to find $(f_i)_{i \in I} \subseteq \mathbb{K}[\Omega]$ such that $\mathbb{K}[f_i : \ i \in I] = \mathbb{K}[\Omega]^G$ holds. To understand the intuition why such a set of generators of the invariant ring is desirable assume that $I = \{1, \dots, m\}$ for some $m \in \mathbb{N}$. In such a case, one can consider the regular function $p : \Omega \to \mathbb{A}^m, \omega \mapsto (f_1(\omega), \dots, f_m(\omega))$. This function detects the clusters of $\bar{\mathsf{C}}$, that is, for all $v, \omega \in \Omega$ we have: $\overline{G\omega} = \overline{Gv}$ iff $p(\omega) = p(v)$. More than detecting the orbit closures could not have been expected in order not to violate the continuity requirements!

If we summarize, desirable properties would be, first, that $I$ is finite and, second, that these functions could be calculated effectively using a 'simple' formula or algorithm.

The first desirable property on finite generation does not hold for arbitrary groups. There is a famous counterexample by Nagata (1959). Adding further assumptions on $G$, namely to be linear reductive, releases more methods from representation theory of groups to apply. In that way, the invariant ring occurs naturally as part of the isotypic decomposition of the regular $G$-representation $\mathbb{K}[\Omega]$ and one can conclude that the invariant ring is finitely generated:

**Theorem 6 (Hilbert (1890), Noether (1915), Hilbert (1933))** *The invariant ring respective to a linear reductive group is finitely generated.*

Regarding the second desirable property, we would like to give a sketch of a constructive proof for finite groups $G$ and $\mathrm{char}(\mathbb{K}) = 0$ by following Noether (1915).

**Proof** We extend the *regular $G$-representation* to a *representation of the group algebra of $G$*, denoted by $\mathbb{K}[G]$, in $\mathrm{End}_{\mathbb{K}}(\mathbb{K}[\Omega])$ induced by the $\mathbb{K}$-algebra morphism

$$\mathrm{H} : \mathbb{K}[G] \to \mathrm{End}_{\mathbb{K}}(\mathbb{K}[\Omega]), \ \alpha \mapsto \left(f \mapsto \sum_{g \in G} \alpha(g) \cdot (g.f)\right).$$

We have $\mathrm{im}(\mathrm{H}(\alpha)) = \mathbb{K}[\Omega]^G$ iff $\alpha \in \mathbb{K}[G]^G$. The constant function $\underline{1} : G \to \mathbb{K}, h \mapsto 1$ is idempotent and spans linear $\mathbb{K}[G]^G = \mathbb{K}.\underline{1}$. Since $\Omega$ is an affine variety, there is a projection of $\mathbb{K}$-algebras $\pi : \mathbb{K}[X_1, \dots, X_n] \to \mathbb{K}[\Omega]$. For $\mu \in \mathbb{N}^n$ define $X^\mu := X_1^{\mu_1} \cdot \dots \cdot X_n^{\mu_n}$, and $|\mu| := \sum_i \mu_i$. Then $\mathbb{K}[\Omega]^G$ is generated by the $J_\mu := \mathrm{H}(\underline{1})\pi(X^\mu)$ for all $\mu \in \mathbb{N}^n$ with $|\mu| \leq |G|$. ∎

## 4. Deep learning classifications

**Definition 7** *Let $n \in \mathbb{N}$. For a subset $X \subseteq \mathbb{R}^n$, we define $\bar{\mathrm{co}}(X) := \{\sum_{i=1}^m \alpha_i x_i \ : \ x_i \in X, m \in \mathbb{N}, \sum_{i=1}^m \alpha_i \leq 1, \alpha_i \geq 0\}$.*

After mathematical modeling, let $\mathsf{C} = (\Omega, (\Omega_l)_{l \in L})$ denote the related classification problem with $\Omega \subseteq \mathbb{R}^n$ compact, and $|L| = p < \infty$. Deep learning provides a computational framework allowing us to find a map $\nu = (\nu_1, \dots, \nu_p)$ given by the composition of $N$ preactivation and activation functions, a so-called $N$-layer feedforward neural network that 'solves $\mathsf{C}$ accurately'. Assume the soft-max score function $\mathbb{R}^p \to \bar{\mathrm{co}}(e_1, \dots, e_p) \subseteq \mathbb{R}^p$ in the

$(N-1)$-th layer, that is $\nu : \Omega \to \bar{\text{co}}(e_1, \ldots, e_p)$, where $e_1, \ldots, e_p$ denote the canonical linear basis of $\mathbb{R}^p$. To formulate precisely, what 'solving C accurately' means, in fact, $\Omega$ must be a probability space, the clusters measurable, and $\nu$ a measurable map!

A resulting $\nu$ is accepted by the majority as an accurate solution if $\text{E}[\nu_m \mid \Omega_l] = \delta_{ml} e_l$ holds for all $m, l = 1, \ldots, p$, where $\text{E}[. \mid .]$ denotes the conditional expectation, and $\delta_{.,.}$ the Kronecker delta. Roughly speaking, when $\nu$ solves C accurately we have $\nu^{-1}(\nu(\omega)) \approx \Omega_l$ for almost all $\omega \in \Omega_l$, for each $l \in L$. In particular, $\nu_i$ is almost constant almost everywhere on $\Omega_l$, for each $l \in L$, for all $i = 1, \ldots, p$.

This consideration shows that when talking about classification, the distribution of data cannot be ignored! In Section 3 the distribution of data was assumed to be uniformly distributed implicitly which can be seen particularly well in the formula of the operator $\frac{1}{|G|} \text{H}(\underline{1})$ used in the proof of Theorem 6.

## 5. Classifications considering data distribution

For all notions concerning functional analysis we refer to Werner (2006), Köthe (1983). For the sake of simplicity, let $G$ be a compact group. Let $L^1(G)$ denote the space of measurable functions on $G$ that are integrable with respect to normalized Haar measure, where, as usual, two such functions are identified if they are equal almost everywhere. The Banach space $L^1(G)$ becomes a Banach *algebra* when equipped with the *convolution product*. We shall require some concepts from the theory of Banach algebras, and refer to Bonsall and Duncan (2012). If $(\Omega, \mathcal{F})$ is a measurable space (where $\mathcal{F}$ is a $\sigma$-algebra of subsets of $\Omega$) and $\Omega$ is a $G$-space, then $(\Omega, \mathcal{F})$ is a measurable $G$-space if the map $(x, \omega) \mapsto x \cdot \omega$ is measurable. To keep it technically as simple as possible, you can think of $\mathcal{F}$ as a Borel $\sigma$-algebra induced by a Fréchet topology.

Let $\Omega$ be a measurable $G$-space and $\mathbb{P}$ be a quasi-invariant probability measure on $\Omega$. For the measurable quotient of $(\Omega, \mathcal{F})$ by $G$, we write $\pi : (\Omega, \mathcal{F}) \to (\Omega/G, \mathcal{F}/G)$.

To understand this section as a continuation of the previous sections, here, we assume that the decomposition of the sample space $\Omega$ into clusters is induced by an action of a compact group $G$ that respects the distribution $\mathbb{P}$ of the data.

As pointed out in Section 4, it is natural to consider neural networks as random variables. Then the 'almost constant almost everywhere on $\Omega_l$ for each $l \in L$' condition translates to the requirement of minimal variance on the clusters.

When starting with an arbitrary random variable $f \in L^1(\Omega, \mathbb{P})$ the canonical way to minimize variances of $f$ on the orbits is given by the conditional expectation with respect to $\pi$. The conditional expectation with respect to the $\sigma$-algebra generated by $\pi$ gives rise to an almost sure unique, idempotent linear operator $\mathbf{E}[? \mid \Omega/G] : L^1(\Omega, \mathbb{P}) \to L^1(\Omega, \mathbb{P})$. By definition, for $\mathbf{E}[f \mid \Omega/G]$ variances on the orbits are minimal, in fact, 0. So we find a linear operator that transforms arbitrary ($L^1$-)random variables to a class of *universal approximators* (Rosenblatt (1958)), the source for functions of our interest as pointed out in Section 3 and Section 4. Unfortunately, the class we end up in this way is given by the simplest feedforward neural networks with just one hidden layer and Binary step as activation function (Figure 1).

If $\mu$ is a measure on the measurable $G$-space $\Omega$ and $g \in G$, then the translate of $\mu$ by $g$ will be denoted by $\mu_g$ and is defined by $\mu_g(E) = \mu(g^{-1} \cdot E)$ for each measurable set $E$

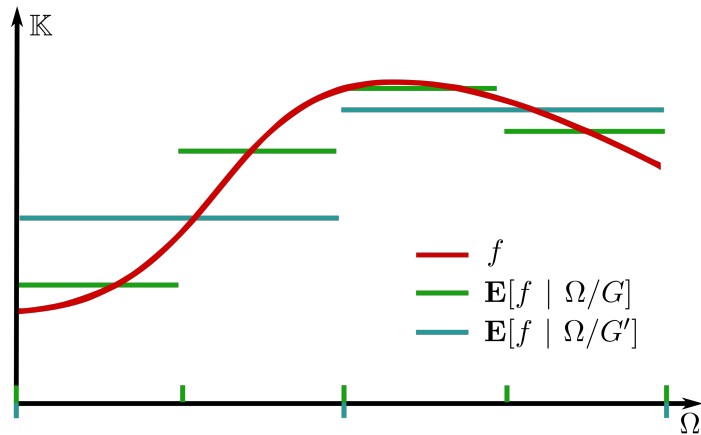

Figure 1: Conditional expectation of $f$ with respect to different group actions $G : \Omega$, $G' : \Omega$ (Wikimedia-Commons).

contained in $\Omega$. Since $\mathbb{P}$ is assumed to be a quasi-invariant probability measure, $L^1(\Omega, \mathbb{P})$ is a Banach left $L^1(G)$-module under the action of $L^1(G)$ defined via the Banach algebra morphism

$$ \text{H} : L^1(G) \to \mathcal{B}(L^1(\Omega, \mathbb{P})), \alpha \mapsto \left( \xi \mapsto \int_G \alpha(g) P(g, ?) \xi(g^{-1} \cdot ?) dg \right), $$

where $\mathcal{B}(L^1(\Omega, \mathbb{P}))$ denotes the bounded linear operators on $L^1(\Omega, \mathbb{P})$ and $P(g, \omega) = (d\mathbb{P}_g/d\mathbb{P})(\omega)$ is the Radon-Nikodym derivative.

As you can see, we have for compact groups as for finite groups (in the proof of Theorem 6) the *regular* representation of the group algebra of the compact group $G$. We can derive this representation even for locally compact groups!

For compact $G$ the idempotent function $\underline{1} : G \to \mathbb{C}, h \mapsto 1$ belongs to $L^1(G)$. Analog, we end up with the projection $\text{H}(\underline{1})$. This is an explicit description of the conditional expectation $\mathbf{E}[? \mid \Omega/G]$!

If we compare the projection in the proof of Theorem 6 with this one we notice that the formula basically differs in $P$. This discussion leads to the conclusion that the cocycle $P$ given by the Radon-Nikodym derivative captures the distribution of the data.

In the non-compact case, say the locally compact case, the constant function $\underline{1}$ is no more available in the domain of H. But we can evaluate $(L^1)$-random variables $u$ of $G$ with expected value 1. They give rise to a new class of functions that are closely related to the original classification problem:

**Definition 8** *Let $G$ be a locally compact group, and $E := \{u \in L^1(G) \; : \; \mathbf{E}[u] = 1\}$. The set of expected-to-be-invariant functions is defined as $\bigcup_{u \in E} \text{im}(\text{H}(u)) \subseteq L^1(\Omega, \mathbb{P})$.*

## 6. Motivation

Our considerations have shown that classification problems induced by group actions are closely related to their associated regular representations of group algebras! Even the invariant ring naturally arises from the representation theory of the regular representation.

Unfortunately, the image of $\mathrm{H}(\underline{1})$ when defined is not well suited for applications since the resulting functions are badly realizable in the application as demonstrated in the discussion.

As is all too often the case, the solution lies in the problem itself. Let us recall these famous examples Figure 2. Or think of smoothly morphing an image into another image. Thus one concludes that in applications the requirement of a disjoint decomposition of the sample space into clusters is unrealistic. And if one thinks about it carefully it is also the reason that the resulting functions $\mathbf{E}[f \mid \Omega/G]$ of the conditional expectation operator are badly realizable in applications. Namely, roughly speaking, the disjoint decomposition causes the 'jumps' in the graph of $\mathbf{E}[f \mid \Omega/G]$.

If one has finitely many classes, say $|\Omega/G| = p \in \mathbb{N}$, one could address this problem by releasing the conditions that is one look at measurable functions $\tilde{\pi} : \Omega \to \partial\bar{\mathrm{co}}(e_1, \ldots, e_p)$ related to the classification problem and consider the conditional expectation with respect to $\tilde{\pi}$. As in the case of $\pi$, a disadvantage is that this operator does not respect the topological/geometric properties of the sample space. These properties of the sample space get essentially lost.

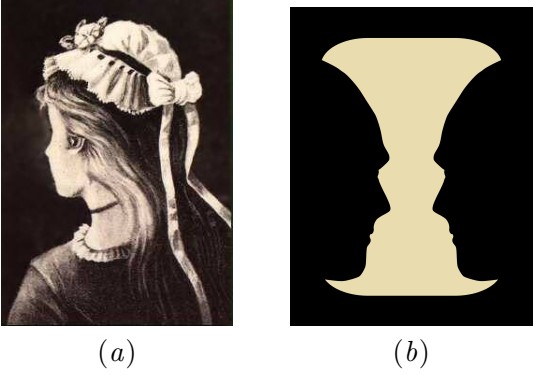

$(a)$        $(b)$

Figure 2: $(a)$ 1888 German postcard: A famous ambiguous image, which can be perceived either as a young girl or an old woman (Anonymous (1888)). $(b)$ An example of Rubin's vase: A famous ambiguous image, which can be interpreted either as two faces or a vase (Rubin (1915), Dilmen).

### 6.1. Probability kernel

Let's assume that the group action is free and have a closer look at the idempotent integral operator H($\underline{1}$). For $f \in L^1(\Omega, \mathbb{P})$, $\omega \in \Omega$ we can rewrite:

$$\begin{aligned} \mathrm{H}(\underline{1})(f)(\omega) &= \int_G P(g, \omega) f(g^{-1} \cdot \omega) dg \\ &= \int_\Omega k(\omega, v) f(v) d\mathbb{P}(v), \end{aligned}$$

where $k : \Omega \times \Omega \to \mathbb{R}$ is defined by

$$k(\omega, v) = \begin{cases} P(g, \omega) & \text{if } \exists\, g \in G \, : \, v = g^{-1}\omega, \\ 0 & \text{else.} \end{cases}$$

The cocycle over the $G$-action $P$ can be interpreted as kind of probability property, that is, $k(\omega, v)$ can be seen as the probability to decide $w$ is related to a given $v$ respectively the occurrence of $v$. Further, H($\underline{1}$) is an idempotent operator, that is $\mathrm{H}(\underline{1})^2 = \mathrm{H}(\underline{1})$ and $P$ satisfies the cocycle identities

$$\begin{aligned} P(gh, \omega) &= P(h, g^{-1} \cdot \omega) P(g, \omega), & (\text{'local'}) \\ P(e, \omega) &= 1 & (g, h \in G, \text{ a.e. } \omega \in \Omega) \ (\text{'normal'}). \end{aligned}$$

These properties translates to

$$\begin{aligned} k(\omega, \omega) &= 1, \\ k(\omega, v) &= \int_\Omega k(\omega, z) k(z, v) d\mathbb{P}(z) \end{aligned}$$

for almost all $\omega, v \in \Omega$. In this way, we end up with an integral operator with respect to a probability kernel that has a density with respect to $\mathbb{P}$.

Thus, the 'local' and 'normal' properties are essential! Now, let's tackle the problem of disjoint clusters without violating the geometry/topology of the sample space. As pointed out in Section 6 disjoint clusters make not so much sense in application-oriented problems. In particular, considering a group action as a starting point of a classification problem makes no sense. We have to switch to the more general notion of a groupoid!

## 7. Groupoids

For all notions concerning category theory and general topology we refer to MacLane (2012) and Köthe (1983).

### 7.1. Definitions and Notation

A group can be understood as a symmetry transformation that relates an object isomorphically to itself (the symmetries of an object, for example, the isometries of a polyhedron), while a groupoid is a collection of symmetry transformations acting between possibly more than one object (nLab authors (2022)).

**Definition 9** *A groupoid $\mathcal{G}$ is a small category in which every morphism is invertible. We denote by $\mathcal{G}$ its set of morphisms, $\mathcal{G}_0$ its set of objects, $s : \mathcal{G} \to \mathcal{G}_0, \gamma \mapsto \gamma^{-1}\gamma$ its source map, and by $r : \mathcal{G} \to \mathcal{G}_0, \gamma \mapsto \gamma\gamma^{-1}$ its range map.*

**Example 1** *Let $Q$ be an arbitrary quiver (see Definition in Appendix B.1). We double the quiver $Q$ by adding to each arrow $\alpha : x \to y$ in $Q$ an arrow $\alpha^* : y \to x$ and obtain the doubled quiver $\bar{Q}$. Requiring that the arrows $\alpha, \alpha^*$ are inverses of each other in the category of paths $\mathcal{P}\bar{Q}$ (see Definition in Appendix B.1) for all $\alpha \in Q_a$ gives rise to a groupoid $\mathcal{G}_Q$.*

**Remark 10** *Let $\mathcal{G}$ be a groupoid. It induces the subset of composable morphisms $\mathcal{G} \times_{s,r} \mathcal{G} \subseteq \mathcal{G} \times \mathcal{G}$ and functions $\mathcal{G} \times_{s,r} \mathcal{G} \to \mathcal{G}, (\gamma, \omega) \mapsto \gamma\omega$, and $\mathcal{G} \to \mathcal{G}, \gamma \mapsto \gamma^{-1}$. These data give rise to a groupoid as Hahn (1978) defined (definition 1.1). And every groupoid in the sense of Hahn (1978) (definition 1.1) gives rise to a groupoid as we defined.*

**Definition 11** *A topological groupoid $\mathcal{G}$ is a groupoid where the set of morphisms $\mathcal{G}$ and $\mathcal{G}_0$ are topological spaces, and all structure maps (source, range, composition, inverse) are continuous maps. Thereby, $\mathcal{G} \times_{s,r} \mathcal{G}$ has the induced topology from $\mathcal{G} \times \mathcal{G}$.*

## 8. Twisted groupoid ∗-algebras

From now on, let $\mathcal{G}$ denote a topological groupoid whose topology is *Hausdorff* and *locally compact*. We also assume that the topology of the groupoid is *second countable*. We now recall the relevant notion from the cohomology theory for groupoids. For a detailed overview see Renault (2006) Section I.1.

**Definition 12** *Let $A$ be a topological abelian group with identity $e_A$. A continuous $A$-valued 2-cocycle on $\mathcal{G}$ is a continuous map $\sigma : \mathcal{G} \times_{s,r} \mathcal{G} \to A$ that satisfies the 2-cocycle identity: $\sigma(\alpha, \beta)\sigma(\alpha\beta, \gamma) = \sigma(\alpha, \beta\gamma)\sigma(\beta, \gamma)$ for all $(\alpha, \beta), (\beta, \gamma) \in \mathcal{G} \times_{s,r} \mathcal{G}$, and is called normalized if $\sigma(r(\gamma), \gamma) = \sigma(\gamma, s(\gamma)) = e_A$ for all $\sigma \in \mathcal{G}$. We write $Z^2(\mathcal{G}, A)$ for the group of continuous $A$-valued 2-cocycles on $\mathcal{G}$.*

We write $\mathbb{T}$ for the multiplicative group of complex numbers of modulus 1. As you can suggest a $\sigma \in Z^2(\mathcal{G}, \mathbb{T})$ will be the counterpart of the cocycle in Section 6. We rediscover the 'local' and 'normal' properties in the definition of the normalized continuous $\mathbb{T}$-valued 2-cocycles. For considering data distribution we need compatible measures. Therefore, we refer to Renault (2006) for the definition of a left Haar system and quasi-invariant measure in this setting. Bourbaki's theory of integration on locally compact spaces is used (Bourbaki (2013), Bourbaki).

We add to the prerequisites that $\mathcal{G}$ is a locally compact groupoid with left Haar system $\{\lambda^u\}_{u \in \mathcal{G}_0}$. We revise our generic definition given at the beginning. The following definition is one of the main achievements of this paper:

**Definition 13** *A classification problem consists of a locally compact, Hausdorff groupoid $\mathcal{G}$ with a left Haar system $\{\lambda^u\}_{u \in \mathcal{G}_0}$, and a normalized continuous $\mathbb{T}$-valued 2-cocycle $\sigma$ on $\mathcal{G}$. The space $\mathcal{G}_0$ is called sample space, and $\sigma$ is called labeling.*

**Remark 14** *Definition 13 covers multi-label classification problems.*

*If the groupoid $\mathcal{G}$ of the classification problem is $\mathcal{G}_Q$ for a quiver $Q$ and the the map $Q_1 \times_{s,r} Q_1 \to Q_0 \times Q_0, (\beta, \alpha) \mapsto (r(\beta), d(\alpha))$ is one-to-one where $Q_1 \times_{s,r} Q_1 \subseteq Q_1 \times Q_1$ denotes the subset of composable arrows. In that case, we would have a single-label classification problem.*

*Under certain conditions, we can consider a spectroid $\mathcal{S}$ of the linearized groupoid $\mathbb{K}\mathcal{G}$ for some field $\mathbb{K}$ (see Gabriel and Roiter (1992) Section 8). In that case, the number of labels relating the samples $v, \omega \in \mathcal{G}_0$ to each other is $\dim \mathcal{R}_{\mathcal{S}}(v, \omega)/\mathcal{R}_{\mathcal{S}}^2(v, \omega)$, where $\mathcal{R}_{\mathcal{S}}$ denotes the radical of $\mathcal{S}$ (see Gabriel and Roiter (1992) Section 3).*

Let $C_c(\mathcal{G})$ denote the locally convex space of complex-valued continuous functions with compact support, endowed with the inductive limit topology (Jarchow (2012), Köthe (1983)).

For $f, g \in C_c(\mathcal{G})$ define $f*g(\alpha) \int f(\alpha\beta)g(\beta^{-1})\sigma(\alpha\beta, \beta^{-1})d\lambda^{s(\alpha)}(\beta), f^*(\alpha) = \overline{f(\alpha^{-1})}\overline{\sigma(\alpha, \alpha^{-1})}$. Under these operations, $C_c(\mathcal{G})$ becomes a topological $*$-algebra (Renault (2006)).

**Definition 15** *This topological $*$-algebra is denoted by $C_c(\mathcal{G}, \sigma)$ and called the $\sigma$-twisted groupoid $*$-algebra of $\mathcal{G}$.*

The $*$-algebras induced by cohomologous 2-cocycles induce isomorphic $*$-algebras.

These $\sigma$-twisted groupoid $*$-algebras are the equivalent counterparts of the $*$-algebras raised in the previous settings. As pointed out before, its regular representation plays an essential role regarding the original classification problem.

### 8.1. Regular representation - Intuitively

The $\sigma$-twisted groupoid $*$-algebra $C_c(\mathcal{G}, \sigma)$ operates on functions on the base $\mathcal{G}_0$. Let $\varphi$ be a function on $\mathcal{G}$, and $u$ a function on $\mathcal{G}_0$. Define

$$(\text{Op}\,\varphi)u(x) := \int_{\alpha:y\to x} \varphi(\alpha)\sigma(\alpha, \alpha^{-1})u(y) \ (...).$$

Intuitively, if we think of the elements of $\mathcal{G}$ as 'arrows' on the base space $\mathcal{G}_0$, then this integral tells us to look at all the arrows $\alpha$ going into a given point $x \in \mathcal{G}_0$, evaluate the function $u$ at the tail of each of those arrows, then move back to $x$ and integrate over all arrows $\alpha$ with 'weight' given by $\varphi$ and labeling given by $\sigma$ (Da Silva and Weinstein (1999)).

## 9. Outlook

The derivation of the Definition 13 suggests that further assumptions on $\mathcal{G}$ like 'linear reductivity' are needed. This will release more methods from representation theory to apply. Simple and indecomposable projective representations will play a central role.

Representation theory of quivers (with relations) can be used in developing and investigating models for classification problems.

The insight of Definition 13 allows us to link unsupervised learning to random walks on the second groupoid cohomology with values in the multiplicative group of complex numbers of modulus 1. Also (model) assumptions in form of Cayley graphs about the second cohomology group are promising for new learning algorithms.

We assume that in applications $\mathcal{G}$ will also be a Lie groupoid. Existing investigations of Lie groupoids could be used in this context.

## Acknowledgments

The author is deeply indebted to his postdoctoral advisers Prof. Dr. Ellen Fritsche, Prof. Dr. Axel Mosig, and Prof. Dr. Markus Reineke for their help, support, kindness, and patience. The author was funded by the European Union's Horizon 2020 Research and Innovation Program, under the Grant Agreement number: 825759 of the ENDpoiNTs project.

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

## Appendix A. Equivariant functions

This section is an appendix for Section 3. As pointed out in Bronstein et al. (2021), besides invariant functions also equivariant functions get into the focus of interest. Let $\chi : G \to \mathbb{G}_m := \mathrm{Gl}_1(\mathbb{K})$ be a character of $G$, that is a group homomorphism. Then, we can define

$$\mathbb{K}[\Omega]^{G,\chi} := \{ f \in \mathbb{K}[\Omega] \; : \; f(g.\omega) = \chi(g)f(\omega) \;\; \forall g \in G, \, \omega \in \Omega \},$$

and $\mathbb{K}[\Omega]_\chi^G := \bigoplus_{n \geq 0} \mathbb{K}[\Omega]^{G,\chi^n}$. By adjusting the sample-space to $\Omega \times \mathbb{A}^1$ and the action of $G$ to $g.(\omega, t) := (g.\omega, \chi(g^{-1})t)$, $g \in G, \omega \in \Omega, t \in \mathbb{A}^1$, we can realize $\mathbb{K}[\Omega]_\chi^G$ as the invariant ring of this adjusted action, that is, we have $\mathbb{K}[\Omega \times \mathbb{A}^1]^G \; \simeq \; \mathbb{K}[\Omega]_\chi^G$.

## Appendix B. Regular representation of twisted groupoid $\ast$-algebras - technically

At first, we need to recall basic notions for algebras, modules, and categories.

## B.1. Algebras, modules

In this section, we will follow closely Section 2 of Gabriel and Roiter (1992). Throughout this section $\mathbb{K}$ denotes a fixed commutative ring.

**Definition 16**

    *i) A $\mathbb{K}$-category is category $\mathcal{A}$ whose morphism sets, here denoted by $\mathrm{Hom}(X,Y)$ or $\mathcal{A}(X,Y)$, are endowed with $\mathbb{K}$-module structures such that the composition maps are $\mathbb{K}$-linear. A $\mathbb{K}$-functor between $\mathbb{K}$-categories $\mathcal{A}$ and $\mathcal{B}$ is a functor $F : \mathcal{A} \to \mathcal{B}$ whose defining maps $F(X,Y) : \mathcal{A}(X,Y) \to \mathcal{B}(FX,FY)$ are $\mathbb{K}$-linear for all $X,Y \in \mathcal{A}$.*

    *ii) A $\mathbb{K}$-category $\mathcal{A}$ is called svelte whose isomorphism classes will form a set.*

    *iii) Let $\mathcal{C}$ be an arbitrary category. By $\mathbb{K}\mathcal{C}$ we denote its linearization, that is the $\mathbb{K}$-category with same objects as $\mathcal{C}$, and its morphism-spaces $\mathbb{K}\mathcal{C}(X,Y)$ are the free $\mathbb{K}$-modules with bases $\mathcal{C}(X,Y)$.*

**Remark 17**

    *i) Each $\mathbb{K}$-algebra $A$ gives rise to a svelte $\mathbb{K}$-category which has one object $\Omega$, and satisfies $\mathrm{Hom}(\Omega, \Omega) = A$. Each svelte $\mathbb{K}$-category $\mathcal{A}$ gives rise to a $\mathbb{K}$-algebra $|\mathcal{A}|$ in the obvious way (unique up to isomorphism of $\mathbb{K}$-algebras). In the sequel, we shall identify svelte $\mathbb{K}$-categories with associated $\mathbb{K}$-algebras, and vice versa.*

    *ii) A functor $E : \mathcal{C} \to \mathcal{A}$ of categories whose range is a $\mathbb{K}$-category uniquely extends to a $\mathbb{K}$-functor $F : \mathbb{K}\mathcal{C} \to \mathcal{A}$.*

**Definition 18 (Gabriel (1972))**

    *i) Let $Q$ be a quiver, that is, a set of vertices connected by arrows as illustrated in Figure 3. We denote by $Q_v$ the set of vertices of $Q$, by $Q_a$ the set of arrows. Thus we have two maps, $d, r : Q_a \to Q_v$ which map an arrow $\alpha$ onto its domain $d(\alpha)$ and its range $r(\alpha)$ ($d(\alpha) = x$, $r(\alpha) = z$ in Figure 3).*

    *ii) A quiver $Q$ gives rise to the category of paths $\mathcal{P}Q$ whose set of objects is $Q_v$, whereas each morphism set $\mathcal{P}Q(x,y)$ consists of the paths with origin $x$ and terminus $y$. The composition is the juxtaposition of paths. We call the linearization $\mathbb{K}\mathcal{P}Q$ of $\mathcal{P}Q$ the $\mathbb{K}$-category of paths of $Q$ and simply denote by $\mathbb{K}Q$.*

The $\mathbb{K}$-category of paths of $Q$ will play an important role in the sequel.

**Definition 19**    *A (left) module $M$ over a $\mathbb{K}$-category $\mathcal{A}$ is a covariant $\mathbb{K}$-functor $M : \mathcal{A} \to \mathbb{K}\text{-Mod}$, where $\mathbb{K}\text{-Mod}$ denotes the $\mathbb{K}$-category of $\mathbb{K}$-modules.*

**Remark 20** *If the $\mathbb{K}$-category $\mathcal{A}$ is svelte, the modules over $\mathcal{A}$ and their morphism again form a $\mathbb{K}$-category which we denote by $\mathcal{A}\text{-Mod}$.*

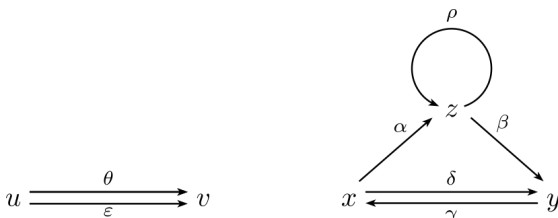

Figure 3: Example of quivers.

**Example 2 (Representation of a quiver)** *A representation of a quiver $Q$ over $\mathbb{K}$ consists of $\mathbb{K}$-modules $V(x)$, one for each $x \in Q_v$, and $\mathbb{K}$-linear maps $V(\alpha) : V(x) \to V(y)$, one for each arrow $(\alpha : x \to y) \in Q_a$. The maps $x \mapsto V(x)$ and $\alpha \mapsto V(\alpha)$ uniquely extend to a $\mathbb{K}$-functor $\mathbb{K}Q \to \mathbb{K}$-Mod, which we still denote by $V$. Thus we identify representation of $Q$ with left modules over $\mathbb{K}Q$. This identification also allows us to transfer the notion of morphism from left modules to representations.*

**Lemma 21** *There is an equivalence of $\mathbb{K}$-categories, induced by the $\mathbb{K}$-functor*

$$| \, . \, | : \mathbb{K}Q\text{-}\mathrm{Mod} \to |\mathbb{K}Q|\text{-}\mathrm{Mod}, V \mapsto \bigoplus_{x \in Q_v} V(x),$$

*where $|\mathbb{K}Q|$-$\mathrm{Mod}$ denotes the $\mathbb{K}$-category of (left) modules over the path algebra of $Q$.*

**Proof** A proof can be found for example in Assem et al. (2006). ∎

### B.2. Representation of $\sigma$-twisted groupoid algebras

The approach from Appendix B.1 will be transferred: The notion of a covariant functor in Definition 19 will be replaced by the notion of a 'twisted functor'. The role of the $\mathbb{K}$-categories will be taken over by $*$-algebras of groupoids and the base category $\mathbb{K}$-Mod will be substituted by the category of separable Hilbert spaces with unitary operators. Thus Renault's *Disintegration Theorem* (Renault, 2006, Theorem II.1.21) becomes the counterpart of Lemma 21 in this setting.

