# OpenReview forum: "On the Ambiguity in Classification"
_NeurIPS.cc/2022/Workshop/NeurReps — NeurReps 2022 Poster_

### Official Review · Reviewer_6dAr · 2022-10-14
**Theoretical and novel paper that would benefit from being more accessible to the ML community**

**Confidence:** 1
**Soundness:** 4
**Presentation:** 3
**Contribution:** 4
**Overall Rating:** 8

**Summary:**

This papers suggests a new mathematical definition of classification that includes ambiguity tasks. They first study the general notion of classification defined with group action and explains the limitations of such a definition. They finally extend this definition with the notion of groupoid.

**Questions:**

Would the authors consider changing 'On the Uncertainty in Classification' to 'On the Ambiguity in Classification'?

How would ambiguous classification differ from multi-label classification (eg: assigning several labels to the same image), and would this lead to a different definition?

**Limitations:**

The authors have not addressed any limitations, and it does not seem to be needed a priori.

**Recommended Decision:**

3: Accept

**Relevance:**

3: Solid fit

**Strengths And Weaknesses:**

It is a well written, significant and novel work on defining ambiguous classification in mathematical terms that would be useful to build up machine learning theory. While being a very technical paper, the authors go straight to the point. It's a novel paper that seems technically sound, even though I am not confident enough to follow the derivations.

A potential issue is that such a technical paper needs an introduction of algebraic geometry and category theory for machine learners. An accessible introduction and a detailed intuition behind the objects used in this paper would be needed. A section dedicated to the notations would be handy.



**Submission Track:**

Proceedings Paper (9 Page)

---

> ### Author Response · Authors · 2022-10-24
> **Responding to your questions**
>
> Thank you for identifying the weaknesses. We will try to revise the points raised.
>
> It is helpful to have the following illustration in mind as a motivation for the definition without labeling:
> > https://www.ihp.fr/sites/default/files/styles/img__1620x1080__image_scale__crop_main/public/media/images/t3-2022.png?h=e02a1d06&itok=2NumuW17
>
> The red dots represent some picked samples, and the edges represent the arrows of $\mathcal{G}$. The heat map on the sample space represents the distribution.
>
> Thank you again for your question. That was a good question. After overthinking, we would like to revise our answer.
> To make it short: Yes, the definition covers multi-label classification.
>
> If the groupoid $\mathcal{G}$ of the classification problem is $\mathcal{G}_Q$ for a quiver $Q$ and the map $Q_1\to Q_0\times Q_0, \alpha\mapsto (r(\alpha), d(\alpha))$ is one-to-one. In that case, we would have a single-label classification problem.
>
> To not dive too much into details, in general, the number of labels depends on the combinatorial invariants of the groupoid. One of them is its `structure quiver'.

---

### Official Review · Reviewer_Ld8G · 2022-10-15

**Confidence:** 1
**Soundness:** 3
**Presentation:** 1
**Contribution:** 2
**Overall Rating:** 5

**Summary:**

The paper provides a theoretical framework to interpret classification problems in the setting of a group acting on an algebraic variety, assuming that the data is uniformly distributed. They generalize this interpretation to the Harmonic Analysis which also takes the data distribution into account. And finally, they define classification problems on a compact groupoid which can also incorporate ambiguous data.

**Questions:**

q1 How is definition 14 useful in building new models for classification or to unify, extend, and theoretically justify an existing class of models?


**Limitations:**

Is there any way to empirically verify that definition 14 is sensible?


**Recommended Decision:**

2: Borderline

**Relevance:**

3: Solid fit

**Strengths And Weaknesses:**

Originality: The contributions are significant and somewhat new.

Quality: The submission is technically sound.

Clarity: The paper is not very clear and some of the claims are difficult to follow. The text is littered with various grammatical errors and typos which makes many of the sentences very hard to understand. It is also written at a very high level of mathematical abstraction without sufficient motivation or explanation.

Significance: The results are somewhat interesting because they generalize classification problems to incorporate ambiguous data.


**Submission Track:**

Proceedings Paper (9 Page)

---

> ### Author Response · Authors · 2022-10-24
> **Responding to your questions**
>
> Thank you for identifying the weaknesses. We will try to revise the points raised.
>
> It is helpful to have the following illustration in mind as a motivation for the definition without labeling:
> > https://www.ihp.fr/sites/default/files/styles/img__1620x1080__image_scale__crop_main/public/media/images/t3-2022.png?h=e02a1d06&itok=2NumuW17
>
> The red dots represent some picked samples, and the edges represent the arrows of $\mathcal{G}$. The heat map on the sample space represents the distribution.
>
> Representation theory of quivers (with relations) can be used in developing and investigating models. The derivation in the paper suggests that simple or projective representations must play a special role. On the other hand, this approach allows a priori assumptions about the sample space when developing models. But also (model) assumptions about the second cohomology group are promising for new learning algorithms.
> We assume that in applications $\mathcal{G}$ will further be a Lie groupoid. Existing investigations of Lie groupoids could be used in this context.

---

### Official Review · Reviewer_WdCt · 2022-10-15
**An abstract definition of classification, albeit unmotivated and unorganized**

**Confidence:** 3
**Soundness:** 3
**Presentation:** 2
**Contribution:** 1
**Overall Rating:** 4

**Summary:**

This paper proposes an abstract definition of classification problem as a tuple of a locally compact groupoid $\mathcal{G}$ (the sample space) and a continuous $\mathbf{T}$-valued 2-cocycle $\sigma$ on $\mathcal{G}$ (the label).

**Questions:**

* For the classification with ambiguous data (like in Figure 2), can the authors expand more on the advantage of their formulation over simply applying the softmax operation and top-k selection?

**Limitations:**

* The authors have not provided the limitations of their framework.

**Recommended Decision:**

2: Borderline

**Relevance:**

3: Solid fit

**Strengths And Weaknesses:**

# Strength

* The authors propose a new framework for classification problems using ideas from $C^*$-algebra. This formulation might have potential to allow geometric ML methods to perform classification tasks with ambiguous labels, which cannot be done with traditional ML methods.

# Weaknesses

* Even with the image morphing example, I think that this article does not have enough motivating examples to demonstrate the importance of this formulation. Geometric ML papers typically give an example of classification on a sphere or on a set of square matrices---maybe the authors can make use of some of these examples.
* Related to the point above, it would be useful to the readers if the authors could provide some experiments on toy data on which this formulation is used.
* In Definition 14, the locally compact groupoid should also be second countable $\mathcal{G}$ to ensure that it has a Haar system [1]
* Figure 1 was taken directly from [Wikipedia](https://en.wikipedia.org/wiki/Conditional_expectation), so the notations in the figure do not match those in the papers. I recommend the authors to make their own figure in order to prevent some confusion.
* The same as above for Figure 3.


References:

[1] Deitmar, A. (2018). On Haar Systems for Groupoids. In Zeitschrift für Analysis und ihre Anwendungen (Vol. 37, Issue 3, pp. 269–275). European Mathematical Society - EMS - Publishing House GmbH. https://doi.org/10.4171/zaa/1613


**Submission Track:**

Proceedings Paper (9 Page)

---

> ### Author Response · Authors · 2022-10-24
> **Responding to your questions**
>
> Thank you for identifying the weaknesses. We will try to revise the points raised.
>
> - If one has finitely many classes (say in the notations of Section 5 one has $\vert \Omega / G \vert = n$), one can also handle ambiguous data in the following way: One defines an at least measurable function $\tilde{\pi}:\Omega \to \partial\bar{\mathrm{co}}(e_1, \ldots, e_n)$
> based on labeled sample data (in the classical sense) and considers the conditional expectation with respect to $\tilde{\pi}$. This operator would also capture approaches taking the softmax operation and top-k selection. The limitations of this are that one assumes a finite number of classes. A disadvantage is that by the map $\tilde{\pi}$ topological/geometric properties of the sample space $\Omega$ are essentially lost.
>
> - At first glance, it looks like [1] can be seen as an example of the suggested framework. However, the details need to be checked.
>
> - In Definition 14 (resp. Definition 13 in the revision) we required that $\mathcal{G}$ has a Haar system.
>
> References:
>
> [1] Richemond, P. H., Dieleman, S., & Doucet, A. (2022). Categorical SDEs with Simplex Diffusion. arXiv. https://doi.org/10.48550/arXiv.2210.14784

---

### Decision · Program_Chairs · 2022-10-21

Accept (Poster)